# Uncovering Abnormal Behavior Patterns from Mobility Trajectories

**DOI:** 10.3390/s21103520

**Published:** 2021-05-19

**Authors:** Hao Wu, Xuehua Tang, Zhongyuan Wang, Nanxi Wang

**Affiliations:** 1National Engineering Research Center for Multimedia Software, School of Computer Science, Wuhan University, Wuhan 430072, China; wuhao961120@whu.edu.cn (H.W.); wzy_hope@163.com (Z.W.); 2018202110014@whu.edu.cn (N.W.); 2School of Remote Sensing and Information Engineering, Wuhan University, Wuhan 430072, China

**Keywords:** mobility trajectory, abnormal behavior pattern, LSTM-based method, spatiotemporal characteristic

## Abstract

Using personal trajectory information to grasp the spatiotemporal laws of dangerous activities to curb the occurrence of criminal acts is a new opportunity and method for security prevention and control. This paper proposes a novel method to discover abnormal behaviors and judge abnormal behavior patterns using mobility trajectory data. Abnormal behavior trajectory refers to the behavior trajectory whose temporal and spatial characteristics are different from normal behavior, and it is an important clue to discover dangerous behavior. Abnormal patterns are the behavior patterns summarized based on the regular characteristics of criminals’ activities, including wandering, scouting, random walk, and trailing. This paper examines the abnormal behavior patterns based on mobility trajectories. A Long Short-Term Memory Network (LSTM)-based method is used to extract personal trajectory features, and the K-means clustering method is applied to extract abnormal trajectories from the trajectory dataset. Based on the characteristics of different abnormal behaviors, the spatio-temporal feature matching method is used to identify the abnormal patterns based on the filtered abnormal trajectories. Experimental results showed that the trajectory-based abnormal behavior discovery method can realize a rapid discovery of abnormal trajectories and effective judgment of abnormal behavior patterns.

## 1. Introduction

The current situation of security prevention and control is extremely severe and normal. Currently, video surveillance methods are mainly used to discover abnormal behaviors and obtain evidence afterwards. Numerous studies on abnormal behaviors have been conducted using video data [1,2,3,4,5,6,7,8,9]. Rai et al. [7] defined the type of motion trajectory and identified the abnormal behavior trajectory by the motion history images and moments. Pathak et al. [8] used probabilistic topic model Probabilistic Latent Semantic Analysis (PLSA) to establish a topic model of local information and quantified the position and size information of the image through rich spatiotemporal gradient descriptors, which extends the topic-based analysis and local descriptors in accuracy. In order to quickly detect violent behaviors in video surveillance, Fillipe et al. [10] used Space-Time Interest Points (STIPs) to classify violence. Tai et al. [11] monitored violent behaviors by calculating optical flow vectors, and Martin et al. [12] used the local multi-scale binary pattern histogram to monitor violent behaviors in real time. However, due to the lack of direct temporal and spatial movement information, traditional security prevention and control technology based on video surveillance plays a very limited role in advanced warning.

The focus of this paper is on the temporal and spatial characteristics of criminal action based on mobile phone trajectory data. Due to the popularity of smart phones, it is possible to extract and discover the laws of personal activities based on mobile phone trajectory data. Using personal spatial activity information to grasp the whereabouts and activities of criminals and curb the occurrence of crimes is a new method for security prevention and control. Based on the supervised or unsupervised methods, different methods are employed to abstract abnormal trajectory from the trajectory dataset [13,14,15]. Yuan et al. [14] determined the similarity between the two trajectories and constructed moving object clustering. Chen et al. [16] proposed a distance function of Edit Distance by Real sequence (EDR), in comparison with DTW, ERP, and LCSS functions. Lee et al. [17] developed the trajectory clustering algorithm TRACLUS to divide a single trajectory into multiple sub-trajectories and used density-based clustering to group them. Tang et al. [18] proposed an SB-DBSCAN-based travel behavior clustering algorithm that combines sampling with density-based clustering to deal with noise in trajectory data. Li et al. [19] proposed an incremental framework to support online incremental clustering. Besser et al. [20] performed distance-based trajectory clustering using new distance measurement. These research results show that it is feasible to conduct abnormal behavior characteristic patterns based on trajectory data. However, the existing research rarely involves the identification of abnormal behavior patterns.

Based on the special regularity of human behavior, many researchers study specific movement patterns based on trajectory data. Some researchers focused on identifying the wandering behavior of elderly people with dementia [21,22,23,24]. Martinez-Ballesté et al. [23] proposed a wandering behavior detection algorithm in outdoor scenes. N.K. Vuong et al. [25] proposed a real-time algorithm to detect the wandering motion. Martino-Saltzman et al. [26] treated travel behavior of nursing home residents as wanderers and non-wanderers. Chang et al. [27] calculated the loss possibility of the elderly by analyzing the area throughout which the pedestrian walks. Di et al. [28] proposed a wandering detection method using the analysis of randomness in patients’ trajectories. Although the content of these studies is different from that of abnormal behaviors in security, these results can provide important method references for studying the characteristics of abnormal behaviors.

This paper mines trajectory data to extract abnormal behavior trajectory and then uncovers abnormal behavior patterns by the characteristic patterns. Figure 1 shows the basic process of the proposed method. First, we preprocessed the collected trajectory data to remove noise. Then, we extracted the relevant trajectory data according to the region of interest. After completing the previous data preprocessing, two core steps were implemented to extract the abnormal behavior trajectory and identify the abnormal behavior type. The moving feature sequences were extracted from trajectory data, and the motion feature of trajectory was constructed by the Long Short-Term Memory network (LSTM) method, then the method based on K-means clustering was applied to distinguish abnormal trajectories. Finally, we defined four patterns of abnormal behaviors of trajectories, including random walk, wandering, scouting, and trailing, which were calculated by the feature matching methods.

The major contributions made by this paper may be summarized as three aspects:(1)This paper implemented the extraction of abnormal behavior trajectory and the identification of abnormal pattern using trajectory data, in terms of the machine learning method and the feature matching method.(2)The abnormal trajectory data were well recognized by LSTM and the K-means cluster method. These methods can achieve data screening in big data without any prior knowledge.(3)The abnormal patterns of social safety were defined, and their feature models were constructed. The abnormal pattern was well identified by the feature matching based on the feature models.

## 2. Research Methods

### 2.1. Data Procession

#### 2.1.1. Noise and Abnormal Points’ Removal

As original trajectory data inevitably had noise and abnormal points, denoising processing was implemented before the identification. Existing denoising processing methods include mean filtering, median filtering, heuristic filtering, Kalman filtering, etc. The mean filtering method judges whether the trajectory Point A is reasonable through the average distance of n points before Point A.The median filter calculates the median distance of the first n points. The common flaw of the above methods is that they have no speed calculation. The Kalman filtering method can reduce the time lag and combine speed and position for the estimation, but the stability of the starting point of the trajectory has a great impact on its subsequent interpolation results. Therefore, this paper developed a heuristic-based outlier detection method to detect and eliminate the noise and abnormal points by the speed and radiation range of each point. The basic steps of this method are as follows:(1)Personal trajectories are roughly inferred based on the normal speed of each person. For example, Bolt’s 100 m race score is 9 s 58, and the speed is about 10 m per second; this value was used as the threshold. If the speed exceeds the threshold, the trajectory point is judged as noise and abnormal points and eliminated.(2)Detect the speed per minute of each point and the corresponding distance radiation of the next minute to determine whether its speed is less than the speed threshold, as shown in Figure 2. If the speed of the track point is greater than the threshold, it is eliminated.

#### 2.1.2. Trajectories of Interest Area Extraction

In real applications, we usually focus on the abnormal behavior of an interest area. Therefore, to improve the processing efficiency, the trajectory data should be firstly filtered out within the interest area. We used the traditional ray algorithm to extract the trajectory points within the interest area. The calculation of the algorithm is shown in Algorithm 1.
**Algorithm 1** Determining if a point is inside a polygon.**Input:** 
$x_{0},y_{0}$: sequence of previously visited locations;**Output:** 
Is point located in polygon1:int crossings = 0;2:**for** (int *i* = 0;*i* < *N* ;*i*++) **do**3: double slope = (y[i + 1] − y[i]) / (x[i + 1] − x[i]);4: bool cond1 = (x[i] ≤ x0) && (x0 < x[i + 1]);5: bool cond2 = (x[i + 1] ≤ x0) && (x0 < x[i]);6: **if** ((cond1 || cond2) && above) **then**7:  crossings ++;8: **end if**9:**end for**10:**if** crossings % 2 == 0 **then**11: return false;12:**else**13: return true;14:**end if**

### 2.2. Initial Screening of Abnormal Trajectories Based on the SeqtoSeq Model

After completing the trajectory screening of the interest area, in order to further narrow the scope of judgment and improve processing efficiency, we used deep learning methods to initially screen out abnormal trajectory data.

Firstly, the mobile feature frequency is extracted from initial trajectory data. An original trajectory $T_{k}$ is represented as:(1)$$T_{k}=\left\{\left(t_{0},lat_{0}lng_{0}\right),\left(t_{1},lat_{1}lng_{1}\right),\cdots ,\left(t_{n},lat_{n}lng_{n}\right)\right\}$$
where $\left(t_{i},lat_{i},lng_{i}\right)$ represents the trajectory point of the pedestrian at time *i* and $lat_{i}$ and $lng_{i}$ are the latitude and longitude coordinates of the pedestrian at time *i*.

To represent more spatiotemporal information of trajectories, $T_{k}^{{}^{\prime }}$ is defined to measure the changes in mobile speed and rotation of trajectory points:(2)$$T_{k}^{{}^{\prime }}=\left\{\left(v_{0},\Delta v_{0},\Delta r_{0}\right),\left(v_{1},\Delta v_{1},\Delta r_{1}\right),\cdots ,\left(v_{n},\Delta v_{n},\Delta r_{n}\right)\right\}$$
where $\Delta v_{i}$ indicates speed change, which is calculated as follows:(3)$$\Delta v_{i}=\left\{\frac{\sqrt{{\left(lat_{i}-lat_{i-1}\right)}^{2}+{\left(lng_{i}-lng_{i-1}\right)}^{2}}}{\Delta t}\right\}$$

$\Delta r_{i}$ represents rotation change, which is calculated as follows:(4)$$\Delta r_{i}=arctan\frac{lng_{i}-lng_{i-1}}{lat_{i}-lat_{i-1}}$$

Then, we abstract the feature sequences $R_{i}$ from the processed trajectory set by a sliding window. The feature sequences $R_{i}$ contain both invariant space and time features, defined as follows:(5)$$R_{i}=\left\{\left(v_{i},\Delta v_{i},\Delta r_{i}\right),\cdots ,\left(v_{i+d},\Delta v_{i+d},\Delta r_{i+d}\right),SF\right\}$$
where *d* is the size of a sliding window and $R_{i}$ is the sliding window that includes $T^{{}^{\prime }}$ ranging from *i* to i+d. SF is the statistic feature set containing 6 statistic parameters. For the sliding window $R_{i}$, SF is defined as below:(6)$$\begin{array}{c} SF=\{\left(v_{i},\cdots ,v_{i+d}\right),(\Delta v_{i},\cdots ,\Delta v_{i+d},\left(\Delta r_{i},\cdots ,\Delta r_{i+d}\right\}\\ \;\times \{mean,max,\frac{3}{4}quantile,\frac{1}{2}quantile,\frac{1}{4}quantile,min\} \end{array}$$

Then, the K-means classification method [29] can be used to extract the abnormal trajectories from the trajectory set. To improve the accuracy of the K-means classification method, based on the SeqtoSeq model [30] (representation learning model), a feature engineering method is applied before using the k-means algorithm to classify normal and abnormal trajectories. Compared with the method of traditional feature engineering, the SeqtoSeq model can transform features of a trajectory into efficient information without labeled data and improve the performance of the K-means algorithm.

The procedure of classification is as follows:

Step 1: Input trajectory feature sequences $h_{k}=R_{0},\cdots ,R_{i}$ into the SeqtoSeq model;

Step 2: Obtain the advanced features $h_{k}^{{}^{\prime }}=f_{SeqtoSeq}\left(h_{k}\right)$ of automatic learning;

Step 3: Input $h_{k}^{{}^{\prime }}$ into the K-means model to cluster the trajectories and extract abnormal trajectory cluster *C*.

Since there are usually more normal trajectories than abnormal trajectories in a trajectory dataset, the number of normal trajectories will be greater than the number of abnormal trajectories after clustering by K-means classification. Therefor, we can extract the abnormal trajectory by judging whether the number of trajectories of each cluster is less than the threshold ϵ:(7)$$Z=\{h_{k}^{{}^{\prime }}\in c^{\left(j\right)}|card\left(c^{\left(j\right)}\right)<\epsilon \}$$
where *Z* is the set of abnormal trajectory ids, $h_{k}^{{}^{\prime }}$ is the high dimension feature of trajectories obtained by the SeqtoSeq model, and $c^{\left(j\right)}$ is the *j*-th cluster obtained by K-means.

### 2.3. Abnormal Patterns’ Identification

After the preliminary screening of abnormal trajectories, the next step is to judge the abnormal type using the mobile feature. We first discuss the types and definition of abnormal behavior, then analyze the characteristics of different types and introduce their judgment methods.

#### 2.3.1. Abnormal Pattern Definition

The motion type embodies the basic characteristics of trajectory movement. The extraction and definition of the motion type initially involve extracting the motion characteristics of the trajectory, which can help us further judge the abnormal patterns. We used the Martino–Saltzman typology method [26] to define the motion types, which is the most widely acknowledged model. Based on this model, the motion types are defined as follows:

**Definition** **1.**
*direct: locomotion from a point to a destination along a straightforward path without significant indecision.*


**Definition** **2.**
*pacing: back and forth locomotion between two points, for which the directional heading is reversed.*


**Definition** **3.**
*lapping: circuitous locomotion revisiting, at least, three points sequentially along the path with several directional changes.*


**Definition** **4.**
*random: locomotion along a haphazard path with multiple changes in direction and several instances of indecision at any point along the path.*

*The motion types represent and define the initial movement characteristics of the trajectory, and the next critical step is to construct the relationship between the motion type and abnormal pattern.*

*In view of the characteristics of common crimes’ preparation [31], abnormal behaviors before a crime mainly include wandering, scouting, random walking, and trailing.*


**Definition** **5.**
*wandering: circuitous locomotion revisiting, at least, three points sequentially along the path with several directional changes.*

*The main feature of wandering behavior is to walk around a place repeatedly. The existing wandering behavior research mainly focuses on the abnormal behavior of patients with Alzheimer’s disease [7,32].*


**Definition** **6.**
*scouting: the same person has at least two similar wandering behavior trajectories.*

*Before committing a crime, most criminals will wander around the crime scene many times to ascertain the situation. Therefore, scouting behavior is the important characteristic of crime preparation, and Chinese law stipulates that scouting a spot is one of the important preparations for crimes. Since scouting behavior often lasts for multiple days, we can define it as a multi-day wandering behavior.*


**Definition** **7.**
*random walking: locomotion along a haphazard path with multiple changes in direction and several instances of indecision at any points along the path.*

*Random crimes are a common type of conventional crime. In random crimes, criminals tend to randomly walk around the crime site, choose the crime site and objects at any time, and commit crimes such as theft.*


**Definition** **8.**
*trailing: two persons take similar trajectories, and one person walks behind another.*

*Criminals usually follow the target person before committing a crime. Trailing behavior means that criminals follow the target person in a certain area, then follow him or on his side. Regardless of whether the target person moves forward or turns, criminals will follow him/her closely. Trailing behavior is an important crime preparation, and Chinese law lists it as one of the crime preparations.*

*Since wandering, scouting, and random walking behaviors can be determined by the characteristics of a single person’s trajectory, we divide them into a large set of categories and discuss their characteristics and judgment methods in the first section. Trailing behavior requires analyzing the trajectory of two people and will be discussed in the next section.*


#### 2.3.2. Wandering, Scouting, and Random Walking Behaviors’ Identification

In order to determine wandering, scouting, and random walking behaviors, we first examined the original trajectory to extract the motion type of the trajectory and further determine the type of abnormal activity pattern. The basic process of identification is shown in Figure 3.

The set of abnormal trajectory ids *Z* screened through Formula (Equation 7) needs to distinguish the trajectory movement behavior first and then further determine the behavior pattern of the trajectory according to the movement behavior.

Because directly using the latitude and longitude information will cause great computational burden to the algorithm, we used a certain division step to divide the sensitive area into a longitude and latitude network GridT, containing M×N small grids, with each small grid $gridT_{i}$ corresponding to an actual geographic area of the sensitive area, where the longitude and latitude of the trajectories are mapped to an M×N matrix.
(8)$$GridT=\{gridT_{0},\cdots ,gridT_{n}\}$$
(9)$$gridT_{i}=\left(x_{i},y_{i},t_{i},lat_{i},lng_{i}\right)$$
where *x*, *y* refer to the horizontal and vertical coordinates of the matrix. $t_{i}$ represents the time of the *i*-th point. $lat_{i}$ and $lng_{i}$ represent the specific longitude and latitude of the *i*-th point.

After mapping the trajectories to the M×N matrix, we assume that if two trajectories’ points fall into the same geographic grid then we consider these two points are equal.
(10)$$gridT_{i}=gridT_{j}\;,if\;\;x_{i}=x_{j}\;and\;y_{i}=y_{j}$$

Next, we used the method proposed in [33] to classify the pattern of the trajectory. N.K. Vuong et al. [33] proposed an algorithm based on a deterministic predefined tree, which performs a state diagram on how the trajectory shifts from direct to random, and then from random to pacing or lapping. The algorithm marks the locations of a trajectory as direct, pacing, or lapping. If a trajectory pattern does not belong to direct, pacing, or lapping, it will be regarded as random. The trajectories processed by this algorithm are based on indoor locations, such as bathrooms and living rooms, while our trajectories are based on a large range of sensitive areas with geographic coordinates. Therefore, we extended the algorithm to meet our studied scenarios, and the concrete procedure of the algorithm is shown in Algorithm 2.

After the movement pattern of the trajectory is recognized, the algorithm will further use the four abnormal behaviors defined above to calculate the true abnormal behavior of the trajectory.

For random walk behavior and direct behavior, since they only involve a single day of motion behavior, they can be directly judged from random walk motion and direct motion as random walking behavior and direct behavior.

For wandering behavior, in order to obtain the abnormal trajectory more accurately, we intended to take advantage of the difference between distance and displacement to judge whether there is wandering behavior. This method enjoys an advantage in speed and can quickly detect specific hovering behaviors in massive trajectory data. The concrete procedure of the algorithm is shown in Algorithm 3.
**Algorithm 2** Identifying mobility patterns.**Input:** 
$GridT_{k}$: sequence of previously visited locations;**Output:** 
pattern type (“direct”, “random”, “lapping”, or “pacing”)1:**if**$\forall (gridT_{a}\neq gridT_{b})$ where a≠b **then**2: label “direct” for the $GridT_{k}$;3:**else**4: Find circles in $GridT_{k}$;5: label pacing for points in the circle whose length is 2;6: label lapping for points in the circle whose length is between 3 and δ;7: **for** each unlabeled sub-sequence $S_{ij}$ of $GridT_{k}$ **do**8:  **if**
$\forall (gridT_{i}\neq gridT_{j})$ where i≠j **then**9:   label “direct” for $S_{ij}$ ;10:  **else**11:   label “random” for $S_{ij}$;12:  **end if**13: **end for**14: $R_{a},L_{a},P_{a}$ = the number of sub-patterns labeled as “random”, “lapping”, and “pacing” respectively;15: $max=Max(R_{a},L_{a},P_{a})$;16: **if**
$max==R_{a}$
**then**17:  label “random” for the $GridT_{k}^{{}^{\prime }}$;18: **else**19:  **if**
$max==L_{a}$
**then**20:   label “lapping” for the $GridT_{k}^{{}^{\prime }}$;21:  **else**22:   label “pacing” for the $GridT_{k}^{{}^{\prime }}$;23:  **end if**24: **end if**25:**end if**

**Algorithm 3** Judge wandering behavior.
**Input:** Divide the trajectory evenly into a area according to time.set threshold :maxscale,vecdis,maxsub.**Output:** the trajectory if has wandering behavior. )1:**for**i=0 to *a* **do**2: Calculate the displacement and distance of the user’s movement in each area;3: **if**
displacement−distance>maxsub
**then**4:  vecdis=vecdis+1;5: **end if**6:
**end for**
7:
**if**
$\frac{vecdis}{a}>maxscale$
**then**
8: return true;9:
**else**
10: return false;11:
**end if**



The distance and displacement in the collected trajectory data were calculated according to Formulas (Equation 11) and (Equation 12). By continuously adjusting the size of the threshold, we selected the threshold that worked best.
(11)$$distance=\sum_{i}^{n-1}\sqrt{{\left(lat_{i}-lat_{i+1}\right)}^{2}+{\left(lat_{i}-lat_{i+1}\right)}^{2})}$$
(12)$$displacement=\sqrt{{\left(lat_{0}-lat_{n}\right)}^{2}+{\left(lat_{0}-lat_{n}\right)}^{2})}$$

For scouting behavior, criminals usually carry out careful preparation before committing crimes in sensitive locations. Knowing the surrounding buildings and police locations in advance can help criminals increase the success rate of a crime. The trajectory of scouting behavior has the following characteristics: long staying time, long moving distance, frequent times. We regulated the scouting behavior based on the following rules.

(1) Long stay time: if the number of staying points of a certain trajectory is far more than the number of staying points of the other normal trajectories, the trajectory is suspected to be scouting behavior.

(2) Frequent and multiple times: if a trajectory has multiple similar trajectories in the sensitive area, the trajectory is suspected to be scouting.

#### 2.3.3. Trailing Recognition Based on the Probability Model

We proposed an effective method to determine the trailing relationship. Our method includes three steps: (1) determining the front–behind relationship; (2) spatiotemporal correlation calculation; (3) discovery of the trailing relationship based on the Gaussian kernel function.

(1)Front–behind relationship determination

We used preliminary screening to filter out unrelated data to improve the speed of the method. By using known trajectory information, the method selects two persons who have traveled with the same range of activities in the same time period in one day.

First, we used linear interpolation to preprocess data to ensure that the time points in the trajectory were spaced evenly and the points with the same index had the same time point. Then, we selected the sub-interval formed by the suspected peer points to calculate the trajectory similarity and calculate the peer relationship.

The meeting point refers to the first index in the trajectory sample dataset that satisfies the corresponding Euclidean distance between the two points that is less than a certain threshold. When Objects A and B are in the same state, the Euclidean distance between each index point will continue to be less than a certain threshold value. Therefore, all points that maintain the peer behavior after the meeting are judged to be suspected peer points.

A one-to-one correspondence is used to find the Euclidean distance between two sample points with the same index of two sample sets. The two points with a distance greater than the threshold maxdist are regarded as separate. When the first distance after the points is less than the threshold mindist, it is considered as a meeting between the two. When the distance between the two is less than maxdist, there is a peer relationship between the two. Thus, a state can be assigned to each trajectory point in the pre-processed dataset.

Because the Euclidean distance only considers the distance relationship when evaluating the relationship between the movement trajectories of two objects, this paper used an additional cosine similarity index to evaluate the similarity of the movement direction points of the object. We then combined the distance relationship and directional relationship to comprehensively analyze trajectory similarity. Because the probability of a trajectory point that exceeds the distance threshold of the suspected peer point is extremely small, the trajectory part containing the suspected peer point can be selected as a sub-interval for measuring the similarity of the trajectory points. After the sub-intervals are set, the cosine similarity between the vectors of two adjacent points is used to determine the credibility of the peers for the sample points existing in the interval.

At a certain point of time *i*, the position point of object $A_{i}$ is $\left(x_{i}^{A},y_{i}^{A}\right)$, and the position point of $B_{i}$ is $\left(x_{i}^{B},y_{i}^{B}\right)$. At the time point i+1, the location point of object A(i+1) is ${\left(x(i+1\right)}^{A}$,$y(i+1{)}^{A})$, and the position of B(i+1) is ${\left(x(i+1\right)}^{B}$,$y(i+1{)}^{B})$. The cosine angle of the two tracks is given by Formula (Equation 13).
(13)$$\cos\;\theta =\frac{\left(A_{i+1}-A_{i}\right)\cdot \left(B_{i+1}-B_{i}\right)}{|A_{i+1}-A_{i}\left|\times \right|B_{i+1}-B_{i}|}$$

The closer the value of cosθ is to 1, the more similar the two vectors are. After traversing all the points to obtain the cosine angle of the n pairs of vectors, we summed them and divided by n to obtain a reference value of the average cosine similarity. According to the average similarity, it was determined whether there was a front–behind relationship.

(2) Spatiotemporal correlation calculation

Based on the trajectory data obtained by the preliminary screening, the known trajectory data of the target and the preliminary screened trajectory data were used to calculate the spatiotemporal correlation degree. We employed the semantic spatiotemporal correlation method to calculate it.

Generally, when criminals intend to trail victims, they may visit a place with variant frequencies at different semantic time periods (random walk, trailing, escape). The trajectories still include those of the criminal and the normal person who passes by the victim briefly. Normal person will not walk randomly so as to find the victim target before trailing. Moreover, a normal person will not escape quickly after trailing, but criminals are just the opposite. Therefore, by using the probability distribution of the visit intensity in different semantic time periods, the normal person can be excluded.

A trajectory is divided into three semantic time periods: random walk, trailing, and escape. After the preprocessing in the previous step, we can obtain the front–behind semantic time period between two trajectories. The time period before the beginning of the front–behind relationship is defined as the random walking period; the time period with the front–behind relationship is defined as the trailing period; and the time period after the end of the relationship is defined as the escape period.

We firstly set a threshold TN and filtered out all elements in the M×N matrix calculated by Formulas (Equation 8) and (Equation 9) that were less than TN to form a region list. The region list of each trajectory at different semantic time periods was ranked in frequency and used to produce the TOP-N region set, which is regarded as:(14)$$T_{s}^{id}=\left\{\left(p_{xy}^{k},x,y\right),k=1,2,\cdots ,N\right\}$$
where $T_{s}^{i}d$ represents the probability distribution at different semantic time periods (random walk, trailing, escape) of a trajectory. $p_{x}y^{k}$ represents the distribution frequency of the trajectory in the k-th region, and (x,y) represents the coordinate of this region.
(15)$$KL\left(T_{s}^{a}∥T_{s}^{b}\right)=\sum_{i\in N}{\left(p_{xy}^{i},x,y\right)}_{a}log\left[\frac{{\left(p_{xy}^{i},x,y\right)}_{a}+\tau }{{\left(p_{xy}^{i},x,y\right)}_{b}+\tau }\right]$$
where τ is a very small value to prevent the denominator from becoming 0.
(16)$$v=\frac{\left(T_{s}^{a}+T_{s}^{b}\right)}{2}$$
(17)$$JSD\left(T_{s}^{a}∥T_{s}^{b}\right)=\frac{\left(KL\left(T_{s}^{a}∥v\right)+KL\left(T_{s}^{b}∥v\right)\right)}{2}$$

After using the same method to obtain the TOP-N region set of Trajectory a and Trajectory b at different semantic time periods (random walk, trailing, escape), we performed the weighting calculation by Jensen–Shannon distances (JSD) to obtain the semantic space–time pattern similarity, which was used to judge whether Trajectory a and Trajectory *b* had the trailing relationship.

(3) Trailing recognition with Gaussian kernel function

After obtaining the spatiotemporal correlation of the trajectory data, the Gaussian kernel function was used to build the discriminate model, providing a list of people who were highly suspected as trailing people along with their corresponding trajectory data.

The Gaussian kernel, also known as the Radial Basis Function (RBF) function, is a commonly used as kernel function. We used the Gaussian kernel function to build the final discriminate model, which maps the finite-dimensional data to a high-dimensional space to obtain a more accurate discrimination result $w_{s}^{a,b}$ using Formula (Equation 18). Then, we determined whether Trajectory *a* and Trajectory *b* had trailing behavior based on whether $w_{s}^{a,b}$ was greater than the threshold λ.
(18)$$w_{s}^{a,b}=\alpha_{1}exp\left(\frac{JSD(T_{s}^{a}∥T_{s}^{b})}{-2\sigma_{1}^{2}}\right)+\alpha_{2}exp\left(\frac{JSD(T_{s}^{a}∥T_{s}^{b})}{-2\sigma_{2}^{2}}\right)$$
where $\alpha_{1}$ and $\alpha_{2}$ are weight coefficients to control the influence of the correlation strength between the criminal and the victim in two spatiotemporal models. $\alpha_{1}$ and $\alpha_{2}$ are the bandwidth of the Gaussian filter, which control the classification strength of the model. The parameter $w_{s}^{a,b}$ defines the possibility of trailing. The threshold of $w_{s}^{a,b}$ can be used to filter out the trajectories and persons who may have a trailing relationship.

## 3. Cases Study and Results

### 3.1. Data Preparation

Since it is difficult to collect the trajectory data of different motion states in the real dataset, in order to verify the performance of the motion state model, we used the simulated dataset and a self-collected dataset to carry out the accuracy experiment of the new method.

The purpose of the simulation dataset D1 was to obtain different types of trajectories according to behavior characteristics. The simulation dataset contained four motion types of simulated trajectories (direct, pacing, lapping, random), each with one thousand data, and a total of four thousand data. In order to increase the authenticity of the simulated dataset, Gaussian white noise was added to each trajectory. Examples of the simulated dataset are shown in Figure 4.

In order to verify the effectiveness of the new method on real data, the second data source D2 was a real trajectory dataset, which was collected by a mobile phone application developed by us. This dataset recorded a total of 6300 trajectories of 35 people for half a year at a time interval of 1 min. Its contents mainly contained the real-time latitude, longitude, and time information of volunteers. First, the collected trajectory dataset was subjected to drift point removal and sensitive area screening. The heat map of the trajectory distribution after data preprocessing is shown in Figure 5. The trajectory distribution on the heat map is sorted by red, yellow, green, and purple in descending order.

As mentioned in Section 2.1, the original trajectory data were preprocessed to remove the noise points and extract the trajectory data within the interest area. Based on the processing results of data preprocessing, the SeqtoSeq feature judgment was performed to initially screen out the trajectory data that may be abnormal using the SeqtoSeq model. The experimental result of the real dataset is shown as Figure 6 and Figure 7, respectively.

In the real dataset, because the user’s trajectory patterns are changeable, it may contain situations other than our defined behavior patterns. If the number of clusters in the k-means algorithm is set to four, the accuracy of the algorithm may drop sharply. Therefore, the elbow joint method was used to determine the trajectory classification number of clusters.

The elbow joint method uses the Sum of Squared Errors (SSEs) to measure the quality of the *k*-means algorithm. The basic idea is that as the number of clusters *k* increases, the sample division will be more accurate, and the degree of aggregation of each cluster will increase, as well as the SSE of data will decrease. When *k* is less than the true number of clusters, the increase of *k* will significantly increase the degree of aggregation of each cluster, so the decline of SSE will be large, and when *k* reaches the number of true clusters, the increase of *k* will not significantly increase the degree of clustering of each cluster, so the decline of SSE will decrease, while the curve will be flat as the number of clusters *k* increases. The inflection point of the change of SSE is what we need. The calculation method of SSE is shown in Formula (Equation 19).
(19)$$SSE=\sum_{i=1}^{k}\sum_{p\in C_{i}}{\left|p-m_{i}\right|}^{2}$$
where $C_{i}$ is the *i*-th cluster, *p* is the sample point in $C_{i}$, and $m_{i}$ is the centroid of $C_{i}$ (the mean value of all samples in $C_{i}$).

In Figure 6, we changed the number of clusters K from one to fifty, and for each K, we calculated the SSE of all data. When the number of clusters *k* was 40, the SSE became stable, so we selected the true number of clusters as 40.

After determining the overall number of clusters, the next step was to determine the number of abnormal clusters in all clusters. The corresponding method was to set the number of clusters from one to the total number of clusters *k* = 40 and select the solution with the best accuracy and time. As shown in Figure 7, if we choose ϵ>30, the accuracy of random walking can reach 100%. However, this did not improve the efficiency of screening abnormal trajectories at all. In order to improve the screening efficiency without losing accuracy, let the number of abnormal categories ϵ be 27.

### 3.2. Motion Type Analysis

In view of the particularity of the motion type, it is difficult to use real dataset to analyze the experimental accuracy of the motion type. In order to solve this problem, the simulation dataset was used to carry out the accuracy analysis experiment of the movement type. In the initial screening procession, the parameters were set as follows: (1) the number of hidden layers was 100; (2) the number of iterations was 300; (3) the learning rate was 0.0001. Based on the initial screening result, we used k-means to cluster the trajectories and examine the motion type. In order to compare the accuracy of the new model with other methods, we compared the experimental results of eight common distance models (SSPD, LCSS, Hausdorff, Frechet, discrete Frechet, ERP, DTW, EDR) with the new method, and the results are shown in Table 1.

In order to better compare the accuracy of the algorithm, the motion type analysis experiment used three evaluation parameters to judge the quality of the algorithm, including precision, recall, and total. Accuracy is the ratio of the recognition result value to its true value in a certain behavior pattern. Its calculation method is as follows:(20)$$Precision=\frac{TP}{TP+FP}$$
where *TP* is the true positive number. FP is the false positive number. The recall rate is defined as: the proportion of the correct judgment of a certain motion type value to the trajectory data value. It is calculated as follows:(21)$$Recall=\frac{TP}{TP+FN}$$
where FN is the false negative number. The total precision is the average of the precision of each type. Its calculation method is as follows:(22)$$Total=\frac{1}{N}\sum Precision$$
where *N* is the number of different class.

As shown in Table 1, the experiment results showed that the overall performance of our method was higher than the performances of the other seven method; however, the recognition accuracy of pacing was low. The clustering algorithm divided the pacing trajectory into the direct type because pacing can be seen as a combination of two direct trajectories. For the real dataset, it was impossible for pedestrians to have no intersection due to the GPS’s low accuracy during pacing. There will be loops in the trajectory, which can avoid identifying pacing as direct.

### 3.3. Effects of Wandering, Scouting, and Random Walking Behavior Identification

In order to verify the recognition accuracy of the new method, this paper used the real trajectory dataset D2 to verify wandering, scouting, and random walking behavior identification. The identification result is showed in Table 2.

Evaluation algorithms usually use two indicators, precision and recall. The experimental results showed that the new method successfully identified most of the abnormal patterns, with an average accuracy of 77%. However, as the accuracy of mobile GPS will have a negative impact on the recognition accuracy, the direct behavior, the wandering behavior, and the random walk behavior may be mistakenly classified as random patterns. The recall rate of the wandering behavior was very high, and the accuracy rate was low. The reason was that the scouting behavior was sometimes mistakenly classified as the wandering behavior. The precision and recall of the random walk behavior were slightly lower because some overlapping patterns were mistakenly classified as random patterns.

Figure 8 shows the part of the experimental results. Figure 8a is an example of the direct behavior, which is identified by the small number of loops and the small number of point intersections in its trajectory. Figure 8b is classified into random walk behavior by the length of the loop in the trajectory, and the moving distance of the trajectory is much larger than Figure 8a,c. Figure 8c shows a typical wandering behavior, which is a very dense loop behavior around a small area. Figure 8d shows an example of scouting behavior. Scouting behavior is a kind of multi-day wandering behavior, as shown in Figure 8(d-1)–(d-4) Among them, the trajectory of each day was a typical wandering behavior. The trajectories belonged to the same person on several days, and we can accurately identify this sequence trajectory as the scouting behavior through the strong similarity of the trajectory.

The definition of abnormal behavior in this paper is a new method, and there is no comparable definition and method. Therefore, it is impossible to conduct precision comparison experiments.

### 3.4. Effects of Trailing Behavior Identification

Since the trailing behavior involves the analysis of the trajectory data of more than two people, it is completely different from the previous analysis steps and methods based on the trajectory data of a single individual. In this paper, in order to judge whether the two trajectories had a trailing behavior, we first judged whether they had front–behind relationship determination, then we judged the similarity between the two trajectories.

The identification results for trailing behavior are shown in Table 3, where σ is the threshold of the JSD model. When the threshold value was 0.75, the precision and recall became the highest. With the increase of the threshold value, the precision and recall gradually decreased. This was because the higher the threshold value was, the lower the fault tolerance rate of the model was, as the two tracks needed higher similarity. In practice, in order to prevent being found, the followers would perform some hidden behaviors, so the higher threshold value was more realistic.

The example of trailing recognition is shown in Figure 9a. Figure 9b shows an example of the trailing behavior with two trajectories. The blue trajectory represents the victim, and the red trajectory represents the criminal. The two trajectories were similar in the same period of time, which can be well separated by our method. Figure 9b shows an example of the trajectories of normal behavior. These two trajectories had no context, and the trajectory intersection points rarely obtained a very low JSD similarity.

## 4. Discussion

### 4.1. Limitations

Although the method we proposed solved the problem of quickly finding abnormal trajectories from trajectory dataset, there are still many limitations in some aspects. First of all, the method proposed in this article is not a parallel algorithm, so when the trajectory data are too many, there will be inefficiencies. Secondly, the algorithm proposed in this paper has defects in trajectory interception. Specifically, we cannot accurately determine the specific time range of the abnormality in the trajectory. A trajectory may only be an abnormal trajectory within 1 h and 30 min, rather than the overall trajectory being an abnormal trajectory. Therefore, it is necessary to accurately split the trajectory data after analyzing the actual collected abnormal data. However, because the number of true abnormal trajectories we collected was too small, the specific split threshold cannot be determined well, so as the number of abnormal trajectories increases, we can analyze and solve this problem.

### 4.2. Conclusions

This paper investigated the relationship between abnormal behavior and personal trajectory data and proposed a new method for identifying abnormal behavior with trajectory data.

Abnormal behaviors were divided into four categories, including wandering, scouting, random walking, and trailing. The abnormal behaviors of wandering, scouting, and random walking were based on the trajectory of a single person, so they were classified into a large set of categories.

For the judgment of the first category of abnormal behavior, the spatiotemporal features of trajectories were extracted based on the SeqtoSeq model, and possible abnormal trajectories were screened out. The motion type was analyzed by the trajectory data that were screened out. Finally, the abnormal patterns of wandering, scouting, and random walking were judged by the motion type and other characteristics.

In order to judge the trailing behavior, the front–behind relationship was determined firstly. Then, the semantic spatiotemporal correlation was calculated and measured the temporal and spatial correlation between the target trajectory and the suspected trailing trajectory. The Gaussian kernel function was used to build the final discriminate model and identify trailing behavior, which mapped the finite-dimensional data to a high-dimensional space to obtain a more accurate discrimination result.

In order to verify the feasibility of the new method, the experiment used two different data sources, namely a simulated dataset and a real dataset. Based on the simulation dataset, the motion state analysis and comparison experiment were carried out. The experimental results showed that our method could obtain better recognition accuracy. Using real datasets, we conducted experiments to discriminate abnormal behavior types. Experimental results showed that our method can accurately and effectively identify the types of abnormal behaviors based on trajectory data.

## Figures and Tables

**Figure 1 sensors-21-03520-f001:**
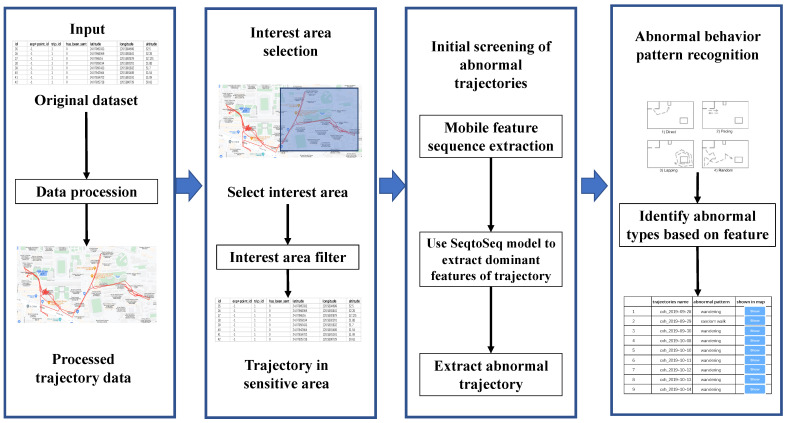
The overall flow of our method.

**Figure 2 sensors-21-03520-f002:**
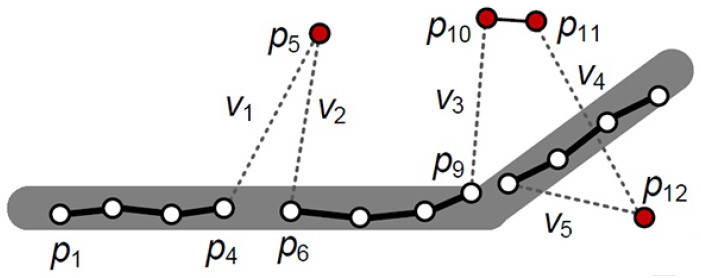
Outlier point removal algorithm.

**Figure 3 sensors-21-03520-f003:**
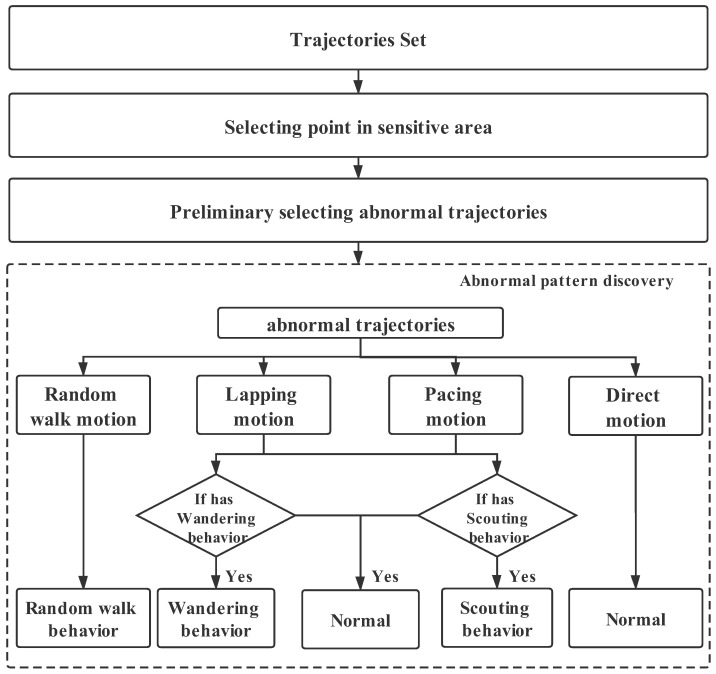
The overview of our identification algorithm.

**Figure 4 sensors-21-03520-f004:**
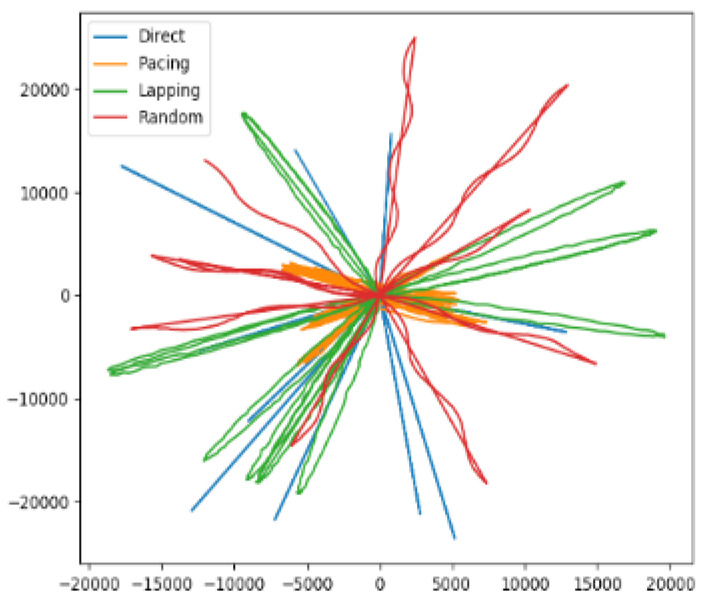
Simulate data of direct, pacing, lapping and random.

**Figure 5 sensors-21-03520-f005:**
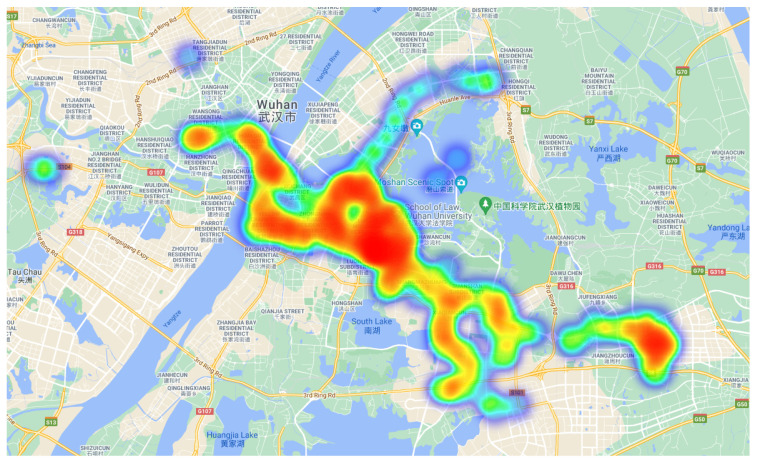
The heat map of the trajectory data in the real dataset.

**Figure 6 sensors-21-03520-f006:**
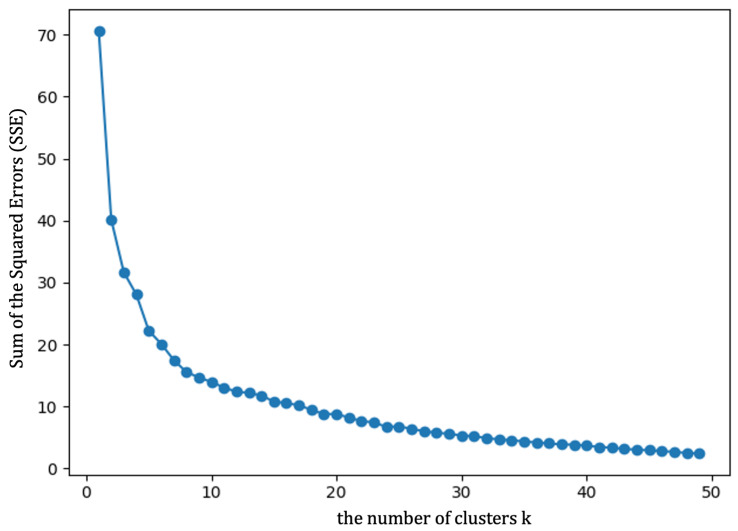
Determination of the cluster number using the elbow method.

**Figure 7 sensors-21-03520-f007:**
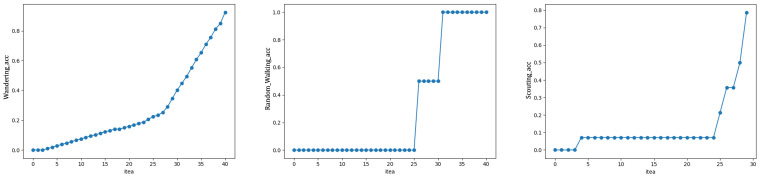
Accuracy comparisons by different thresholds ϵ.

**Figure 8 sensors-21-03520-f008:**
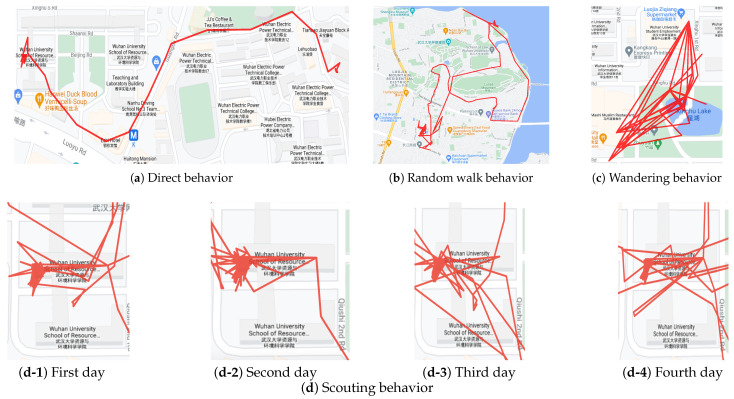
Examples of random, wandering, and scouting behavior identification.

**Figure 9 sensors-21-03520-f009:**
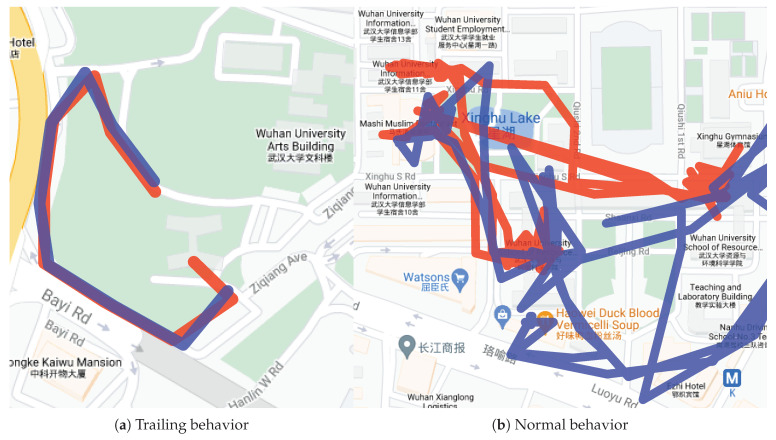
The example of trailing recognition.

**Table 1 sensors-21-03520-t001:** Clustering performance on the simulated dataset.

Result	Accuracy (Precision/Recall)				
	Direct	Pacing	Lapping	Random	Total
SSPD	0.46/0.60	0.40/0.40	0.40/0.40	0.41/0.50	0.47
LCSS	0.37/0.60	0.25/0.40	0.31/0.50	0.25/0.40	0.47
Hausdorff	0.45/0.50	0.35/0.50	0.21/0.37	0.35/0.30	0.45
Frechet	0.75/0.30	0.31/0.50	0.35/0.60	0.25/0.40	0.45
Discrete Frechet	0.71/0.50	0.35/0.50	0.35/0.50	0.28/0.40	0.47
ERP	0.31/0.50	1.00/0.40	0.54/0.60	0.31/0.50	0.50
DTW	0.33/0.30	0.30/0.40	0.38/0.50	0.44/0.40	0.40
EDR	0.26/0.50	0.47/0.90	0.31/0.60	0.36/0.70	0.67
Our Method	1.00/0.87	0.89/0.98	0.99/0.98	0.98/1.00	0.96

**Table 2 sensors-21-03520-t002:** Identification results of mobility behavior patterns for a single person.

Result	Accuracy			
	Direct	Wandering	Scouting	Random Walk
Precision	0.88	0.77	0.73	0.66
Recall	0.44	0.98	0.48	0.50

**Table 3 sensors-21-03520-t003:** Identification results of the trailing patterns.

σ	Accuracy	
	Precision	Recall
0.75	0.88	0.77
0.80	0.86	0.74
0.85	0.83	0.70
0.90	0.78	0.65

## Data Availability

The data presented in this study are available upon request from the corresponding author.

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
