# Peer review of "Uncovering Abnormal Behavior Patterns from Mobility Trajectories"

_sensors, 2021, doi:10.3390/s21103520_

Round 1

Reviewer 1 Report

1.The quality of some Figures in the paper (Figure 8, 9, etc.) should be strengthened.

2.The author emphasized that this paper proposes a novel method to discover abnormal behaviors and judge abnormal behavior patterns using mobility trajectory data.  Therefore, I think the author should strengthen the discussion of the proposed method, the comparison with the previous methods and the advantages of the method in this paper.

L9 and L71: Long Short Term Mermory network(LSTM)

L25, L28: PLSA and STIP should also be given their full names when they first appear.

L72-74: Check the writing, it is not a complete sentence.

L101: '16-18m/s per second' /s and per second are repeated.

I think the author should strengthen the discussion of the proposed method, the comparison with the previous methods and the advantages of the method in this paper. For example, the author may refer to the following literature.

Dong-Her Shih, Ming-Hung Shih, David C. Yen & Jia-Hung Hsu (2016) Personal mobility pattern mining and anomaly detection in the GPS era, Cartography and Geographic Information Science, 43:1, 55-67, DOI: 10.1080/15230406.2015.1010585

BAO, P. M., JI, G. L., WANG, C. L., & ZHU, Y. B. (2017). Algorithms for mining human spatial-temporal behavior pattern from mobile phone trajectories. DEStech Transactions on Computer Science and Engineering, (cst).

Author Response

Thanks very much for taking your time to review this manuscript. I really appreciate all your comments and suggestions! Please find my itemized responses in below and my revisions/corrections in the re-submitted files.

1.The quality of some Figures in the paper (Figure 8, 9, etc.) should be strengthened.

Response:

Thank you for your suggestion. I replaced the pictures in the paper with high-definition pictures and modified some unclear pictures.

2.The author emphasized that this paper proposes a novel method to discover abnormal behaviors and judge abnormal behavior patterns using mobility trajectory data.  Therefore, I think the author should strengthen the discussion of the proposed method, the comparison with the previous methods and the advantages of the method in this paper.

L9 and L71: Long Short Term Mermory network(LSTM)

L25, L28: PLSA and STIP should also be given their full names when they first appear.

L72-74: Check the writing, it is not a complete sentence.

L101: '16-18m/s per second' /s and per second are repeated.

Response:

Thank you for your suggestions. I have revised the above questions in the paper. I marked the modified position in green.

I think the author should strengthen the discussion of the proposed method, the comparison with the previous methods and the advantages of the method in this paper. For example, the author may refer to the following literature.

Dong-Her Shih, Ming-Hung Shih, David C. Yen & Jia-Hung Hsu (2016) Personal mobility pattern mining and anomaly detection in the GPS era, Cartography and Geographic Information Science, 43:1, 55-67, DOI: 10.1080/15230406.2015.1010585

BAO, P. M., JI, G. L., WANG, C. L., & ZHU, Y. B. (2017). Algorithms for mining human spatial-temporal behavior pattern from mobile phone trajectories. DEStech Transactions on Computer Science and Engineering, (cst).

Response:

This article did not conduct a comparative experiment with the above-mentioned related research mainly based on the following two considerations:

(1) The current research on anomaly detection is mainly divided into two aspects: 1. Some behavior patterns of a person are different from normal people. These problems are that normal trajectories and abnormal trajectories can only be distinguished through similar clustering methods, but the specific abnormal behaviors of abnormal trajectories cannot be specifically identified. 2. A person's behavior pattern is different from one's normal behavior pattern. These abnormal trajectories are compagreen with normal trajectories, and the algorithm cannot specifically calculate the specific abnormalities of the trajectory. The above two types of research are only to judge whether an abnormal situation occurs in a trajectory, and cannot specifically identify criminal acts. Many trajectories, but these abnormal trajectory discrimination algorithms do not specifically identify the type of abnormal trajectory. Not only that, many experiments do not disclose the correct detailed code and distinguish the abnormal behavior defined in this article. Therefore, it cannot be compagreen with our algorithm. This paper defines a new abnormal behavior pattern based on the four patterns contained in a trajectory, and recognizes the trajectory in the article.

(2) The behavior patterns of the two trailing trajectories will recognize that the two trajectories are normal trajectories according to the existing clustering method or semantic similarity. However, in the real scene, the two trajectories have a high degree of coincidence, and the trailing behavior defined in this article has a unique context. The existing algorithms do not identify this type of abnormal behavior.

Reviewer 2 Report

Review for paper titled “Uncovering Abnormal Behavior Patterns from Mobility Trajectories”.

This paper examines abnormal and normal behaviour patterns of mobility using k-means clustering.

First, thank you for the opportunity to review this paper.

I want to begin by saying I thoroughly enjoyed this article. It is well presented and on an interesting topic. I found the introduction to be well structured and clear in defining the remit of the article. I do have some suggestions for improving the manuscript. In general, at the moment it reads more as a descriptive account of what was done, but there is little in terms of why the work was done and why certain decisions were made.

In light of this, I have attached a series of comments which I hope are beneficial to the authors.

Major considerations:

  1. The article needs a thorough proof-read for spelling and grammar.
  2. I am interested in the elimination of datapoints based on normal speeds (line 100). In the example given, the normal car speed is quite a narrow range (16-18m/s). Where did these criteria come from? Is this a valid and fair criterion to apply? I think elaboration on all such criteria is important with justification for each.
  3. There is a lack of justification for many critical decisions. For example in section 2.3.1. there are definitions of wandering, scouting, etc. The authors note on line 417 that the definition of abnormal behaviour in this paper is a new method. They therefore needs to be very clear on how they have come to their definition within this article. Why, for example, is wandering defined and operationalized as it is in this paper? This applies throughout the paper, I continuously found myself asking “why” did the authors do what they did, of “why” do they define things as they do, or “why” is their decision justified.
  4. More detailed information is needed from lines 362-364 in terms of choosing how many clusters to use. How was 30 decided upon?
  5. The authors need to include a section at the end of the article which explains how this could be used in the real world. The authors have used pre-existing data, but this is collected after abnormal behaviour occurs. While I have an idea of the potential for this, the authors should be more explicit on exactly how what they have done has a real world implication. For example, could it be used in some way to aid prevent criminal behaviour? How would this work exactly?
  6. A limitations section is also important to qualify the exact remit of this work and what can and cannot be taken from the project at its current stage. This should then be used to explain potential directions for future work.

Minor considerations

  1. Line 4: a space is missing between a period and the word “Abnormal”. This occurs again on line 406, and 408.
  2. For readability of the text, I suggest the authors be consistent in how they cite and include author names. For example, on line 57 the authors say “Martino-Saltzman et al. [17] treated travel…” and this, in my opinion is great as I get to see the authors and it reads logically, because people don’t really “read” what is in the []. However in many instances, such as line 55, the authors say “[18] proposed a wandering behavior…”. It would be so much easier to read if author names were included in these types of sentences, so in this case saying “Martinez-Ballesté et al. [18] proposed a wandering…” would be better.

Author Response

Thanks very much for taking your time to review this manuscript. I really appreciate all your comments and suggestions! Please find my itemized responses in below and my revisions/corrections in the re-submitted files.

Major considerations:

  1. The article needs a thorough proof-read for spelling and grammar.

Response:

Thank you for your suggestion. I have corrected the spelling and grammatical errors in the full text. I changed the position and marked it in green.

  1. I am interested in the elimination of datapoints based on normal speeds (line 100). In the example given, the normal car speed is quite a narrow range (16-18m/s). Where did these criteria come from? Is this a valid and fair criterion to apply? I think elaboration on all such criteria is important with justification for each.

Response:

Thank you very much for your comments. For your question, the 16-18m/s in the article is an example, and it is not used in the paper. In the paper, we use 10m/s as the threshold of the drift point removal algorithm. The detailed calculation of the threshold and the reason for the selection have also been explained in the paper. Generally speaking, Bolt's 100m score is 9s58, and his average speed is 10.4m/s. The speed of human movement cannot exceed this speed. Therefore, if the speed of two points in the trajectory data is greater than the speed of 10m/s in the collected trajectory data, we consider these two points as abnormal points which need to be removed.

  1. There is a lack of justification for many critical decisions. For example in section 2.3.1. there are definitions of wandering, scouting, etc. The authors note on line 417 that the definition of abnormal behaviour in this paper is a new method. They therefore needs to be very clear on how they have come to their definition within this article. Why, for example, is wandering defined and operationalized as it is in this paper? This applies throughout the paper, I continuously found myself asking “why” did the authors do what they did, of “why” do they define things as they do, or “why” is their decision justified.

Response:

We first find that m defines 4 people's normal behavior patterns in the thesis, which are direct, lapping, pacing and random walk. Since everyone has the above-mentioned behavior patterns, it is necessary to extract abnormal behaviors from the above-mentioned behavior patterns.

According to the article [John S Strahorn Jr. “Preparation for crime as a criminal attempt”. In:Wash. & Lee L. Rev.1 (1939)], the criminal will prepare for the crime before committing the crime. Generally speaking, it can be found from the synthesis of many crime processes that criminal suspects usually wander, step on, follow, and walk randomly before committing a crime. However, there is no related article on these criminal behaviors that have defined and specifically identified where the abnormal trajectory of the pedestrian trajectory appears. Based on this, we innovatively propose four abnormal behaviors and calculate them.

The definition of the trajectory in our paper is based on the analysis of the trajectory data in the Huawei campus. However, due to Chinese laws and Huawei regulations, the trajectory data is in a private state, so we cannot analyze the real abnormal trajectory data in this paper, but can only explain the definition of abnormal behavior.

In addition, the thresholds of the papers and implemented codes in this article cannot be explained using real dataset containing abnormal trajectories, so we can only use the real data sets collected by ourselves to compare and verify the methods in this paper. Wandering behavior has related work [E. Batista, F. Casino, and A. Solanas. “On wandering detection methods in context-aware scenarios”. In:2016 7thInternational Conference on Information, Intelligence, Systems Applications (IISA). 2016, pp. 1–6.], but the work is for the elderly, so we solve the problem by changing the threshold and condition. For stepping behavior, we believe that if a person has stepping behavior, he must wander around a place many times, and in order to avoid being discovegreen, the interval of him is relatively long. What we detect here is to judge every other day. If it is recognized as wandering behavior under different days, and judged according to the trajectory similarity algorithm, it is judged that the trajectory is indeed the trajectory of the same person (that is, the trajectory is relatively similar), then it is proved to have stepping on behavior. Random walk behavior is also expanded according to m's definition of random walk movement in the paper. The definition of trailing behavior is the behavior between two people, and it is also the behavior that often occurs before the crime. This is our innovative proposal. Because the previous definition did not involve the behavior pattern between the two trajectories, and the trailing behavior often appeagreen before the crime preparation. Therefore, we analyze its related characteristics and understand that the two trailing trajectories have a front-behind relationship, and the trajectories of the trailing person and the trailed person have a high degree of consistency when the trailing behavior occurs.

  1. More detailed information is needed from lines 362-364 in terms of choosing how many clusters to use. How was 30 decided upon?

Response:

In response to your question, I have explained the selection of clustering points in the paper in detail.The elbow joint method uses Sum of the Squagreen Errors(SSE) to measure the quality of the k-means algorithm. The basic idea is that as the number of clusters k increases, the sample division will be more accurate, and the degree of aggregation of each cluster will increase, the  of data will decrease. When k is less than the true number of clusters, the increase of k will significantly increase the degree of aggregation of each cluster, so the decline of  will be large, and when k reaches the number of true clusters, the increase of k will not significantly increase the degree of clustering of each cluster, so the decline of  will decrease, and the curve will be flat as the number of clusters k increases. The inflection point of the change of  is the clustering value we need. The calculation method of  is shown in the formula 19.

  1. The authors need to include a section at the end of the article which explains how this could be used in the real world. The authors have used pre-existing data, but this is collected after abnormal behavior occurs. While I have an idea of the potential for this, the authors should be more explicit on exactly how what they have done has a real world implication. For example, could it be used in some way to aid prevent criminal behavior? How would this work exactly?

Response:

The algorithm shown in this article has been used in real scenarios, but due to our specific system application and Huawei signed a confidentiality agreement, the specific use process of the software cannot be described in detail, but the specific process of the specific algorithm is shown in Figure 1 in the paper. First of all, the trajectory data will be collected every day. then, the collected data will be preprocessed and added to the system's database, so that the database will be continuously expanded and improved, which can continuously improve the accuracy of the algorithm's judgment. When a new trajectory is input, seqtoseq and k-means are used to initially screen out abnormal trajectories, and the behavior trajectory of the trajectory is judged for classification processing. next,we use the algorithm in 2.3.2 and 2.3.3 to determine the type of abnormality of the trajectory.

  1. A limitations section is also important to qualify the exact remit of this work and what can and cannot be taken from the project at its current stage. This should then be used to explain potential directions for future work.

Response:

In response to your question, we have already added it to the paper. We found two limitations while implementing the algorithm

  1. the method proposed in this article is not a parallel algorithm, so when the trajectory data is too large, there will be inefficiencies.

2.Secondly, the algorithm proposed in this paper has defects in trajectory interception. Specifically, we cannot accurately determine the specific time range of the abnormality in the trajectory. A trajectory may only be an abnormal trajectory within 1 hour and 30 minutes, rather than the overall trajectory being an abnormal trajectory. Therefore, it is necessary to accurately split the trajectory data after analyzing the actual collected abnormal data.However, because the number of true abnormal trajectories we collected is too small, the specific split threshold cannot be determined well, so as the number of abnormal trajectories increases, we can analyze and solve this problem.

Minor considerations

  1. Line 4: a space is missing between a period and the word “Abnormal”. This occurs again on line 406, and 408.

Response:

Thank you for your suggestions. I have revised the above questions in the paper. I marked the modified position in green.

2.For readability of the text, I suggest the authors be consistent in how they cite and include author names. For example, on line 57 the authors say “Martino-Saltzman et al. [17] treated travel…” and this, in my opinion is great as I get to see the authors and it reads logically, because people don’t really “read” what is in the []. However in many instances, such as line 55, the authors say “[18] proposed a wandering behavior…”. It would be so much easier to read if author names were included in these types of sentences, so in this case saying “Martinez-Ballesté et al. [18] proposed a wandering…” would be better.

response:

Response:

Thank you for your suggestions. I have revised the above questions in the paper. I marked the modified position in green.

Round 2

Reviewer 1 Report

The author replies or revises the questions well